# The Putative Auto-Inhibitory Domain of Durum Wheat Catalase (TdCAT1) Positively Regulates Bacteria Cells in Response to Different Stress Conditions

**DOI:** 10.3390/antiox11091820

**Published:** 2022-09-15

**Authors:** Mouna Ghorbel, Kaouthar Feki, Sana Tounsi, Nouha Bouali, Malek Besbes, Faiçal Brini

**Affiliations:** 1Department of Biology, College of Sciences, University of Hail, P.O. Box 2440, Ha’il City 81451, Saudi Arabia; 2Laboratory of Biotechnology and Plant Improvement, Center of Biotechnology of Sfax, P.O. Box 1177, Sfax 3018, Tunisia

**Keywords:** abiotic stress, bioinformatic analysis, durum wheat, *E. coli*, growth inhibition, catalase, protein expression

## Abstract

Catalase is a crucial enzyme in the antioxidant defense system protecting organisms from oxidative stress. Proteins of this kind play important roles in controlling plant response to biotic and abiotic stresses by catalyzing the decomposition of H_2_O_2_. The durum wheat catalase 1, TdCAT1, has been previously isolated and characterized. Here, using bio-informatic analysis, we showed that durum wheat catalase 1 TdCAT1 harbors different novel conserved domains. In addition, TdCAT1 contains various phosphorylation residues and S-Nitrosylation residues located at different positions along the protein sequence. TdCAT1 activity decreased after treatment with λ−phosphatase. On the other hand, we showed that durum wheat catalase 1 (TdCAT1) exhibits a low CAT activity in vitro, whereas a deleted form of TdCAT1 has better activity compared to the full-length protein (TdCAT460), suggesting that TdCAT1 could present a putative autoinhibitory domain in its C-terminal portion. Moreover, we showed that TdCAT1 positively regulates *E. coli* cells in response to salt, ionic and osmotic stresses as well as heavy metal stress in solid and liquid mediums. Such effects had not been reported and lead us to suggest that the durum wheat catalase 1 TdCAT1 protein could play a positive role in response to a wide array of abiotic stress conditions.

## 1. Introduction

Plant growth is affected by different environmental stresses such as drought, light variations, salinity, extreme temperatures and osmotic stress. All of these factors limit worldwide agricultural production as well as the survival of susceptible genotypes. Plant exposure to these stresses causes the production of reactive oxygen species (ROS, e.g., OH^−^, O_2_, H_2_O_2_, ^1^O_2_). ROS act as signaling molecules in cells, but they are toxic at high concentrations [1]. ROS have specific properties (sites of production, levels of reactivity and ability to cross biological membranes) [1,2]. They are produced in peroxisomes, mitochondria, chloroplasts, and in any other cellular compartments that present biological compounds with a high redox potential [3]. ROS are detoxified by an important antioxidative system [3,4], including catalases (CAT), superoxide dismutase (SOD), ascorbate peroxidases (APX) glutathione/thioredoxin (Trx h) and glutathione sulfo-transferases (GST).

CATs, iron porphyrin enzymes, are present in all aerobic organisms. These metalloenzymes are highly conserved and represent the major H_2_O_2_ scavenging enzyme [5]. Moreover, they are regulators of different biological and physiological processes such as growth and development, cell cycle and photosynthetic functions as well as plant responses to biotic and abiotic stresses [5,6,7]. In plants, CATs are encoded by a multigenic family [8]. The number of genes encoding for CAT proteins can vary from one gene in sweet potato (*Ipomoea batatas* L.) [9], tomato (*Lycopersicon esculentum* Mill.) [10] and castor bean (*Ricinus communis* L.) [11] to fourteen in the *Brassica napus* genome [12]. Expression of CAT genes is regulated temporally and spatially and controls plant growth and maturation as well as plants’ response to environmental stresses [13,14]. In *Gossypium hirsutum* L. and *G. barbadense* L., all seven CAT genes encoding for catalase proteins were induced after plant infection with *V. dahlia* Kleb pathogen [8]. In *Brassica napus*, BnCAT1–BnCAT3 and BnCAT11–BnCAT13 genes were significantly upregulated after application of different stresses such as salinity, abscisic acid (ABA) and cold and gibberellic acid (GA), whereas most genes, except BnCAT6-9 and 10, were responsive to waterlogging stress [12].

In monocotyledons, little is known about molecular regulation of catalases or their classification. The first identified catalases in plants were isolated from rice and maize. In fact, it has been shown that ZmCAT2 and CATC isolated from *Zea mays* and *Oryza sativa*, respectively, could be implicated in H_2_O_2_ scavenging during photorespiration [15]. Later, ZmCAT1 was isolated, and it was shown that this oxydo-reductase was implicated in glyoxysomal activity [16]. Some other catalase genes were isolated from other monocotyledons such as *Hordeum vulgare* [17], *Zantedeschia aethiopica* [18], *Musa accuminata* [19], *Triticum aestivum* [20] and *Triticum monococcum* [21].

We have previously isolated the first catalase gene from durum wheat (TdCAT1) [22], which belongs to the subfamily 1 of the catalase proteins. The TdCAT1 protein is located in the peroxisome, but the deletion of the C-terminal part of the protein containing the peroxisomal targeting signal (PTS1) causes a cytoplasmic location of this protein [21]. TdCAT1 was upregulated in wheat after application of different abiotic stresses (NaCl, PEG, H_2_O_2_ and MnCl_2_) [22]. Moreover, TdCAT1 confers abiotic stress tolerance to yeast [22]. Recently, it has been demonstrated that TdCAT1 harbors a conserved Calmodulin- binding domain located at its C-terminal portion [23]. This domain interacts with Calmodulins (CaM) in a calcium-independent manner but stimulates the catalytic activity of the protein in a calcium-dependent manner. Moreover, TdCAT1 harbors several conserved cation-binding domains located in different parts of the protein that stimulate its activity [23]. In this work, we performed bioinformatic analysis of TdCAT1. Moreover, we characterized the importance of the domain (400–460) as a possible autoinhibitory domain of TdCAT1, involved in the regulation of the catalase activity of the protein controlling the tolerance of bacteria against various stresses. As far as we know, this is the first report that describes the role of the autoinhibitory domain of catalase proteins in response to abiotic stresses in bacteria.

## 2. Materials and Methods

### 2.1. Bioinformatic Analyses

The Netphos database (https://services.healthtech.dtu.dk/service.php?NetPhos-3.1, accessed on 4 April 2022) was used to identify the putative phosphorylation sites in TdCAT1. The presence of Signal sequences and location of their cleavage sites were predicted through SignalIP-5.0 server (http://www.cbs.dtu.dk/services/SignalP/, accessed on 4 April 2022). Moreover, the http://smart.embl-heidelberg.de/ program and the https://www.ebi.ac.uk/interpro/soft-ware (accessed on 4 April 2022) were used to identify the functional domains in TdCAT1 structure. Multiple sequence alignments of catalase proteins sequences were performed using Multalin software (http://multalin.toulouse.inra.fr/multalin/ (accessed on 4 April 2022), accessed on 6 April 2022). The https://www.ebi.ac.uk/interpro/software (accessed on 6 April 2022) was used to identify the functional domains in TdCAT1 structure. The SOPMA database was used to generate the 2D dimensional model and the number of β sheets and α helices. The HMMTOP database (http://www.enzim.hu/hmmtop/html/document.html, accessed on 6 April 2022) was used to identify the presence of transmembrane helices in TdCAT1. The Signal IP-5.0 database (https://services.healthtech.dtu.dk/service.php?SignalP-5.0, accessed on 7 April 2022) was used to search for cleavage sites in TdCAT1. 

### 2.2. Production of Full-Length and Truncated Forms of TdCAT1 and Their Derivative Forms in E. coli 

To produce the recombinant proteins His_TdCAT1 and the different truncated forms [His_TdCAT_200_ (1-200aa); His_TdCAT_295_ (1-295aa); His_TdCAT_340_ (1-340aa); His_TdCAT_400_ (1-400aa) and His_TdCAT_460_ (1-460aa)], the corresponding cDNA of each protein type was amplified by PCR with the Pfu Taq DNA polymerase and, using the appropriate primers (Appendix A), digested by the appropriate restriction enzymes, *EcoR*I and *Xho*I, and cloned in-frame with Histidine-tag into the pET28a expression vectors (Novagen, Madison, WI, USA) *EcoR*I and *Xho*I restriction sites. The resulting constructs pHis_TdCAT1, pHis_TdCAT_200_, pHis_TdCAT_295_, pHis_TdCAT_340_, pHis_TdCAT_400_ and pHis_TdCAT_460_ were introduced into the BL21 (DE3) *E. coli* strain (Novagen). A single selected colony from each construction was grown overnight at 37 °C in LB medium containing 100 µg/mL Kanamycin, with shaking at 220 rpm. The culture was next diluted 1:100 into fresh LB-Kanamycin medium and grown to an OD of 0.6 at 600 nm. Protein expression was then induced by 1 mM isopropyl β-d-thiogalactopyranoside (IPTG) overnight at 37 °C. 

To purify the recombinant proteins, bacterial cells were harvested by centrifugation at 4500× *g* rpm for 10 min at 4 °C, and the pellets were subsequently washed twice with cold water. Later, the cells were harvested in cold lysis buffer (Tris-HCl 100 mM pH 8; EDTA 1 mM; NaCl 120 mM; 1mM DTT, 50 mM PMSF and 0.5% Tween) and sonicated on ice. Afterwards, the cells were centrifuged at 9000× *g* rpm for 45 min at 4 °C. The deleted forms were purified from the supernatant, but pHis_TdCAT1 was not found in the supernatant. Thus, the recovered inclusion bodies were resuspended and incubated overnight in lysis buffer at 4 °C with agitation, then centrifuged at 9000× *g* rpm at 4 °C for 10 min. The supernatant was then loaded on a Ni-Sepharose column (Bio-Rad, California, USA), pre-equilibrate with binding buffer (Tris-HCl 100 mM pH 8; NaCl 0.5 M; 30 mM imidazole) and gravity eluted. Protein quantification was performed using the Bradford method [24], and the correct size of recombinant proteins was checked by SDS-PAGE electrophoresis.

To investigate bacterial tolerance to abiotic stress, the overnight cultures (BL21 (DE3) *E. coli* strain) were inoculated into fresh LB medium (1:100 dilution). A total of 100 μg/mL kanamycin was added to the medium, and then the solution was incubated for 4 h at 37 °C until the exponential growth phase (OD600 = 0.5 − 0.6). IPTG was added to cultures to a final concentration of 1 mM, which were incubated at 37 °C for up to 4 h to induce expression of the inserted genes. After that, cells were subjected to different abiotic stress conditions (200 mM NaCl, 1 mM CaCl_2_, 400 mM Sorbitol and 200 mM KCl) and metallic stress conditions (750 μM CuCl_2_, 0.5M LiCl, 750 μM CdCl_2_, 750 μM ZnCl_2_, 750 μM Fe-SO_4_ and 750 μM AlCl_3_), as previously described [21,25,26]. These assays were performed in the presence of control assays without stresses. For drop assays, a serial dilution (from 10^−1^ to 10^−6^) was performed, and 5 μL of each dilution was spotted on different mediums containing or not containing the appropriate stress. Plates were then incubated at 37 °C for 15 h, and the results were analyzed.

For LB liquid medium assays, 500 μL of induced cell cultures was diluted in 15 mL fresh LB medium and the growth pattern of control as well as stress-treated cells was noted for 26 h by taking OD at 600 nm every 2 h time point. The data obtained in triplicates were averaged and used to plot the graph.

### 2.3. CAT Activity Assays

CAT activity was determined according to Aebi, [27] with minor modification. CAT activity was measured spectrophotometrically at 240 nm using a specific absorption coefficient at 0.0392 cm^2^μmol^−1^ H_2_O_2_. One mL of substrate solution made up of 50 mM H_2_O_2_ in a 75 mM phosphate buffer at pH 7.0 and 160 µg of proteins were mixed at 25 °C for 1 min, and the reaction was stopped by adding 0.2 mL of 1 M HCl. The enzyme activity was assayed from the rate of H_2_O_2_ decomposition as measured by the decrease in absorbance at 240 nm and was calculated as μmol H_2_O_2_ decomposed/mg protein/min.

### 2.4. Phosphatase Treatments

To generate a dephosphorylated His-TdCAT1 (dpHis-TdCAT1) protein, 500 µg of the purified protein was treated with 1200 Units of λ−phosphatase (New England Biolabs, Ipswich, MA, USA) at 30 °C for 30 min in a buffer containing 50 mM HEPES, 10 mM NaCl, 2mM DTT, and in presence of 1 mM Mn^2+^. The reaction was stopped by heating at 65 °C for 15 min in the presence of 3 mM Na_2_EDTA.

### 2.5. Viability Test

A viability test was performed using 50 μL of liquid assay, spaced on LB basal plate, then incubated at 37 °C overnight. The results were observed by counting the number of colonies present on the medium.

### 2.6. Statistical Analysis

Each experiment was carried out in at least four replicates. Student’s t-test was performed to determine significant differences between the means of bacterial growth in unstressed and stressed mediums. The percentage presented in the following figures was calculated by the data of survival of bacteria transformed by empty pET-28a or pET-28a_TdCAT1 or deleted forms, grown in LB medium supplemented or not supplemented with the mentioned stress. The results were compared statistically by bacteria grown in unstressed medium and differences were considered significant at *p* < 0.01. Mean values that were significantly different at *p* < 0.01 from each other are marked with asterisks (*).

## 3. Results

### 3.1. In silico Sequence Analyses of TdCA1 Protein

Analysis of TdCAT1 structure using the SMART database (http://smart.embl-heidelberg.de/, accessed on 4 April 2022) showed that TdCAT1 contains a catalase domain (PF00199; 18–401 aa) and catalase-related domain (PF06628; 423–448 aa). This confirmed the conserved domain structure in the TdCAT1 protein (Figure 1a). Moreover, InterPro database (https://www.ebi.ac.uk/interpro/, accessed on 6 April 2022) revealed the presence of two conserved domains in TdCAT1 structure: a catalase proximal active site signature domain (54–70 aa) and a catalase proximal heme-ligand signature (344–352 aa). These domains are conserved in many catalases from monocotyledonous and dicotyledonous plants, such as *Triticum monococcum* (TmCAT1) [21] and *Panax ginseng* (PgCAT1) [25]), and several other plant species (Figure 1b,c). In addition, using the transmembrane helix prediction (HMMTOP) database, no transmembrane helices were identified in TdCAT1, which suggested that this oxydo-reductase has no function on the cells’ membranes. The same result was shown for some other proteins such as PgCAT1 [25] and TmCAT1 [21]. Furthermore, TdCAT1 does not present any signal sequences and presents no cleavage sites, as revealed by the SignalIP-5.0 server (http://www.cbs.dtu.dk/services/SignalP/, accessed on 4 April 2022). Secondary structure analysis and molecular modeling of TdCAT1were conducted by the SOPMA program. The secondary structure analysis revealed that TdCAT1 consists of 135 α-helices, 27 β-turns jointed by 74 extended strands, and 256 random coils (Appendix A). This is highly similar to the secondary structure of CAT1s from *Brachypodium dictachyon* (BdCAT1) catalase protein, which contains 134 α-helices, 29 β-turns jointed by 72 extended strands, and 257 random coils and Alpine snowbell (*S. alpina*), which contains 124 α-helices, 38 β-turns jointed by 77 extended strands and 253 random coils (Appendix A). In a second step, TdCAT1 structure was analyzed using NetPhos3.1 database. Using this server, we identified a total of 41 potential phosphorylation sites in TdCAT1 sequence (Appendix A) and 4 sites of Nitrosylation (Appendix A). 

Recently, it has been shown that the bovine catalase has a S-nitrosylated cysteine located at the Cys 377 in the LGPNYLQIPVNCPYR motif [28]. Interestingly, no cysteine is detected in the TdCAT1 sequence, but the S-nitrosylated cysteine residues are present at 85, 230, 325 and 470 (Appendix A). The same observation is also available for the other studied catalases in this work, except for C470, which is present only in durum wheat TdCAT1 and absent in the other studied catalases (data not shown).

### 3.2. Role of CAT1 Phosphorylation on the Activity of TdCAT1

Protein phosphorylation is crucial to enhance the catalytic activity of different proteins such as TMPK1 [29,30]. To evaluate if TdCAT1 phosphorylation is critical for its activity, His-TdCAT1 was treated with λ-phosphatase prior to in vitro catalase assays. Results showed that the V0 of the dephosphorylated form (dpHis-TdCAT1) decreases by about 34% compared with the native form, His-TdCAT1 (Figure 2a). This V0 decrease is caused by TdCAT1 dephosphorylation and not by the heat treatment of the protein at 65 °C for 15 min used to stop the λ-phosphatase reaction. In fact, the V0 of catalase reaction remains unchanged when the same heat treatment was applied to His-TdCAT1 (Figure 2a). A heat-treated λ-phosphatase was also used to dephosphorylate His-TdCAT1. As shown in Figure 2, the phosphatase activity of His-TdCAT1 was unchanged after incubation with inactivated λ-phosphatase. These findings support that TdCAT1 might be phosphorylated in *E. coli* by endogenous protein kinases. Moreover, TdCAT1 phosphorylation stimulates the catalase activity of TdCAT1. As catalase activity of TdCAT1 was stimulated by cations, especially Fe^2+^ and Mn^2+^, we added 2 mM of each cation (Ca^2+^, Mg^2+^, Zn^2+^, Cu^2+^, Fe^2+^ and Mn^2+^) to the medium after λ-phosphatase treatment. As shown in Figure 2b, the presence of the divalent cations did not modulate the catalytic activity of TdCAT1. These results suggest that the phosphorylation of the catalase is crucial for cation stimulation of its activity. 

### 3.3. Mapping of a Putative Autoinhibitory Domain in the TdCAT1 Sequence

To analyze the catalase activity of the different deleted forms, all the constructions were produced and then purified, and the correct size of the constructions was verified (Figure 3a,b). Furthermore, we verified that all mutated forms produced the same amount of proteins, as revealed by Western blot analysis (Figure 3b). The optimum buffer concentration, pH and temperature for TdCAT1 proved to be 75 mM, 7 and 25 °C, respectively [23]. Thus, we determined the initial reaction rate (Vo) of TdCAT1 by measuring enzyme kinetics of the purified recombinant deleted proteins during the first minute. Our results confirmed TdCAT_200_ to have a very low catalase activity, as previously shown [23]. Moreover, the catalase activity of TdCAT295 and TdCAT340 forms was lower than that of the complete TdCAT1 (Figure 3c), suggesting that the presence of the whole catalase domain of TdCAT1 is essential for the protein’s activity. Interestingly, the catalase activity of TdCAT_400_ was 1.24-fold higher than TdCAT1 activity in normal assay conditions. This finding suggested that the C-terminal portion of the enzyme exerts a possible constitutive inhibitory effect on the enzyme activity. Such a domain should be localized in the C-terminal portion of the protein, as the same catalase activity of TdCAT_460_ level was equal to that of the non-truncated protein (Figure 3c).

The elimination of a 60 aa fragment (between 400 and 460aa) located at the C-terminal portion of TdCAT1 resulted in a slight increase of the enzyme of basal activity. These findings suggested that the 60 residues between TdCAT_400_ andTdCAT_460_ could play a role in the inhibition. The low activity of TdCAT_340_ could be explained by the fact that TdCAT1 needs the presence of the whole catalase domain to exert its activity. Taken together, our results suggested that wheat catalase TdCAT1 harbors a previously unidentified autoinhibitory domain, spanning 400 to 460 aa residues, that regulates TdCAT1 activity.

### 3.4. Recombinant TdCAT1 Proteins Confer Bacterial Tolerance to Ionic and Osmotic Stress

The biological role of TdCAT1 in abiotic stress tolerance was investigated using heterologous expression in *E. coli* cells. The full-length cDNA of TdCAT1 and its derivatives forms were cloned in the pET128a expression vector, then transformed into *E. coli* (BL21 strain). The growth of transformed *E. coli* cells transformed with TdCAT1—the deleted forms, or the empty vector—was determined under different stresses (LB media containing or not containing 400 mM Sorbitol; CaCl_2_ 10 mM; 200 mM NaCl, cold stress (4 °C, 3 h) and 300 mM KCl) in solid and liquid mediums. Under standard conditions, there was no significant growth difference between all tested strains, and all strains grew equivalently in solid mediums (Figure 4a). Instead, under stress treatments, cells transformed with TdCAT1 recombinant plasmids exhibited greater growth rates in comparison with cells transformed with empty vector for all stress treatments (Figure 4b–e), except for sorbitol (Figure 4f). In the latter case, bacteria transformed with the empty vector and with all the deleted TdCAT1 forms were more tolerant to sorbitol compared with TdCAT_200_. In fact, in all tested conditions, TdCAT_200_ transformed bacteria were very sensitive and exhibited a low multiplication rate compared with the other strains. This indicates that the presence of the totality of the catalytic site is crucial for the activity of TdCAT1.

The same result was also observed for bacteria grown on liquid mediums (Figure 5). In fact, bacteria were growing at an equivalent rate in the absence of stresses (Figure 5a). By contrast, in the presence of the mentioned stresses, TdCAT1 and the derivative forms confirmed tolerance to NaCl, KCl, Sorbitol and cold stress (Figure 5b–d). The percentage of viable cells was also investigated. The number of the recombinant cells was almost the same for bacteria grown under normal conditions (Figure 6a), about 1.5-fold higher under sorbitol stress (Figure 6b), 1.8-fold higher under NaCl stress and KCl (Figure 6c,d) and 2-fold higher under cold stress (Figure 6e). These results indicated that the expression of TdCAT1 in *E. coli* cells has a positive effect on their growth under different stress conditions.

### 3.5. TdCAT1 Positively Regulates Bacterial Response to Heavy Metal Stress of E. coli Cells

The growth of *E. coli* cells containing pET28a-TdCAT1 vectors, the TdCAT1 deleted forms or the empty vector pET28a was analyzed in media alternatively containing CdCl_2_, AlCl_3_, LiCl, CuCl_2_, ZnCl_2_ and FeSO_4_ (Figure 7). In drop assays and under normal conditions, the growth pattern of recombinant cells was similar to the growth of control cells (Figure 7a). However, cells transformed with pET28a-TdCAT1 and all the deleted forms, except for TdCAT_200_, exhibited better growth in comparison with cells transformed with empty vector in medium containing all these heavy metals treatments (Figure 7b–e). These results suggested that TdCAT1 could play a potential role in heavy metal stress tolerance. The same results were observed for bacteria grown on liquid media (Figure 8). All bacteria grew at the same rate in the absence of stress (Figure 8a), whereas in the presence of the mentioned stresses, TdCAT1 and the derivative forms confirmed tolerance to all metals tested (Figure 8b–d). The percentage of viable cells was also investigated: the number of the recombinant cells was almost the same for bacteria grown under normal conditions (Figure 9a), whereas it was about 2.25-fold higher under LiCl stress (Figure 9b), 1.8-fold higher under AlCl_3_, FeSO_4_ and CuCl_2_ stress (Figure 9c–e) and 2.3-fold higher under CdCl_2_ stress (Figure 9f). These results indicated that the expression of TdCAT1 in *E. coli* cells has a positive effect on their growth under different heavy metals stress conditions.

## 4. Discussion

Abiotic stresses like salinity, drought and heavy metals are fatal threats to agriculture that cause enormous yield losses [31]. Drought and salinity are two of the most common abiotic stresses in dry and arid regions. In these regions, plants have developed different physiological mechanisms to grow under difficult climatic conditions [27,31,32]. Thus, comprehension of the mechanisms of plant tolerance to abiotic stress can help to develop stress-tolerant crops [30,31,32]. Abiotic stresses activate several genes, such as those for catalase enzymes, which are implicated in H_2_O_2_ scavenging [12]. One TdCAT1 gene was identified in the durum wheat genome that belongs to class I. This tetrameric enzyme confers tolerance to different abiotic stresses in yeast and transgenic *Arabidopsis* [22]. TdCAT1 harbors a PTS domain that controls a peroxisomal localization of TdCAT1, and the suppression of this motif causes a cytoplasmic localization of those proteins [21]; but few bio-informatic analyses are available concerning this protein. In this report, we performed bioinformatic analysis to study the characteristics of TdCAT1. Our results showed that the TdCAT1 protein harbors 41 potential phosphorylation sites. Treatment of TdCAT1 with λ-phosphatase causes the decrease of the catalytic activity of about 35%. Moreover, addition of divalent cations (Ca^2+^, Mg^2+^, Zn^2+^, Cu^2+^, Fe^2+^ and Mn^2+^) stimulates the catalytic activity of native TdCAT1, as previously shown [23], but not the dephosphorylated protein dpTdCAT1. This could reveal the importance of phosphorylation in regulating TdCAT1 in planta. One *TdCAT1* gene was identified in the durum wheat genome that belongs to class I catalases. This tetrameric enzyme confers tolerance to different abiotic stresses in yeast and transgenic *Arabidopsis* [22]. The TdCAT1 protein harbors a PTS domain that controls its peroxisomal localization, and suppression of this motif causes a cytoplasmic localization of such proteins [21]. In this report, we performed bioinformatic analyses to study the characteristics of TdCAT1. Our results showed that TdCAT1 protein harbors 41 potential phosphorylation sites. Treatment of TdCAT1 with λ-phosphatase causes the decrease of the catalytic activity, whereas addition of divalent cations (Ca2+, Mg2+, Zn2+, Cu2+, Fe2+ and Mn2+) stimulates the catalytic activity of native TdCAT1 (as previously shown [23]) but not that of the dephosphorylated protein, dpTdCAT1. This is suggestive of the importance of phosphorylation in regulating TdCAT1 in planta. On the other hand, we found that the TdCAT1 sequence presents four different S-nitrosylated cysteine residues, as revealed by the GPS-SNO 1.0 database. Nitric oxide (NO)-controlled post-translational modifications (PTMs) are important for cell signal transduction under normal and stressful conditions, as well as during different metabolic and physiological conditions. Among others, S-nitrosylation is the most investigated protein PTM; it occurs during protein signaling, synthesis, or degradation to maintain cell homeostasis. S-nitrosylation mediates hormone signaling in plants. NO targets Cys residue(s) of proteins involved in hormonal signaling and regulates their functions [33]. S-nitrosylation controls reactive nitrogen species (RNS) and ROS homeostasis in plants via catalase proteins. S-nitrosylation is regulated by a group of proteins termed transnitrosylase [34]. Recently, it has been demonstrated that Arabidopsis CAT3 is required for S-nitrosylation. CAT3 presented a transnitrosylase activity at Cys-10 [35]. This activity is regulated by a highly conserved residue (Cyc-343) that is not conserved in CAT2 (the enzyme responsible for the majority of catalase activity in Arabidopsis) [35].

It has been recently shown that bovine and pepper catalases present a S-nitrosylated cysteine residue at Cys 377 and Cys370, respectively [28]. Interestingly, this residue is not conserved in all the catalases studied in this work. In contrast, other sites were identified in TdCAT1: C85, C230, C325 and C470. Of them, whereas C85, C230, and C325 residues are conserved in all the catalases studied, the C470 residue was found to be present in the TdCAT1 sequence only. These dissimilarities between the studied catalases suggest that further investigations are needed to establish where the TdCAT1 can be nitrosylated in durum wheat. Point mutation of those residues could help to clarify their role in vivo.

NO controls the signaling pathway of all hormones (auxin, abscisic acid, ethylene, jasmonic acid, brassinosteroid, gibberellic acid, salicylic acid, cytokinin, and strigolactone). Many proteins have been identified as S-nitrosylation targets during hormonal signaling, such as TIR1 in *A. thaliana* and *Triticum aestivum* L., implicated in auxin signaling [36,37,38], BIN2 in *Zea mays*, implicated in Brassinosteroid signaling to positively regulate oxidative stress in plants [38] and MYB30 in *A. thaliana*, activated during abscisic acid signaling [39,40]. Such findings strongly confirm the role of catalase proteins in phytohormone signaling.

To study the catalytic activity of different protein portions, we have generated deleted forms of TdCAT1. After expression of the different fragments in *E. coli* and catalase assay of the corresponding proteins, we found that the first 200 aa of TdCAT1 has a very low catalase activity in vitro. This could be explained by the absence of the whole catalase domain in this fragment. Similarly, TdCAT295 and TdCAT340 presented a low activity compared with the full length, but this activity was more pronounced compared with TdCAT200. This suggested that the whole catalase domain is important for a full catalase activity of TdCAT1. On the other hand, the TdCAT400 form presented an increased activity compared to the full length TdCAT1. Thus, we speculated that TdCAT1 presents a putative autoinhibitory domain, spanning about 60 aa (residues 400–460). Such a finding was not previously reported and may suggest another regulatory mechanism of TdCAT1 in planta, based on the autoregulatory domain. The presence of autoinhibitory domains was reported in many proteins, such as phosphatases like PP5, Sox-2 and Sox-11 [41], WNK [42], CAX1, an *Arabidopsis* Ca2+/H+ Antiporter [43], wheat Mitogen Activated protein Kinase Phosphatase 1, TMKP1 [44] and Protein Kinase 5 [45].

It has been extensively reported that the growth of *E. coli* cells can be regulated under different stress-expressing recombinant proteins [46,47,48]. Moreover, sugarcane catalase ScCAT1 was shown to enhance *E. coli* cells’ growth under different abiotic stress conditions [32]. To gain insight into the role of TdCAT1 in response to different abiotic stresses, we overexpressed the protein in *E. coli* cells exposed to different stresses. We showed that TdCAT1 acts as a positive regulator of bacterial response to all abiotic stress tested, such as salt and heavy metals (AlCl_3_, CuCl_2_, CdCl_2_, ZnSO_4_ and FeSO_4_). A positive role in regulating plants’ response to abiotic stresses was previously described for TdCAT1 [22]. In fact, overexpression of TdCAT1 in *Arabidopsis* enhances plants’ tolerance to many stresses, such as salt, PEG and MnCl_2_. Moreover, the same result was also described in yeast overexpressing TdCAT1 [22]. Taken together, these results suggest that TdCAT1 is functional and plays a crucial role in bacterial response to these abiotic stresses.

## 5. Conclusions

CAT, SOD and POD proteins are involved in protecting cells against oxidative stress. In this work, we characterized the durum wheat *TdCAT1* gene in vitro by heterologous expression in *E. coli* cells. As far as we know, this is the first report of a wheat CAT gene that describes its role in regulating bacterial defense against abiotic stress in vitro. Moreover, TdCAT1 protein harbors several putative phosphorylation sites and a putative autoinhibitory domain that slightly inhibits the catalase activity of TdCAT1. This domain could play a crucial role in regulating TdCAT1 activity in vivo. Further studies on TdCAT1 and on the role of the putative autoinhibitory domain as well as on its regulation under normal and stressful environments in planta are needed to better understand TdCAT1 structural–functional characteristics. Moreover, the presence of four different S-Nitrosylated residues was described here. Such residues were not described before in monocotyledons and were reported in bovine catalase and *C. annuum*. The data presented in this work will be useful for a better understand of catalase proteins and their implication in controlling ROS scavenging mechanisms in plants. In conclusion, TdCAT1 gene could have a promising role for the development of crops under different abiotic stresses.

## Figures and Tables

**Figure 1 antioxidants-11-01820-f001:**
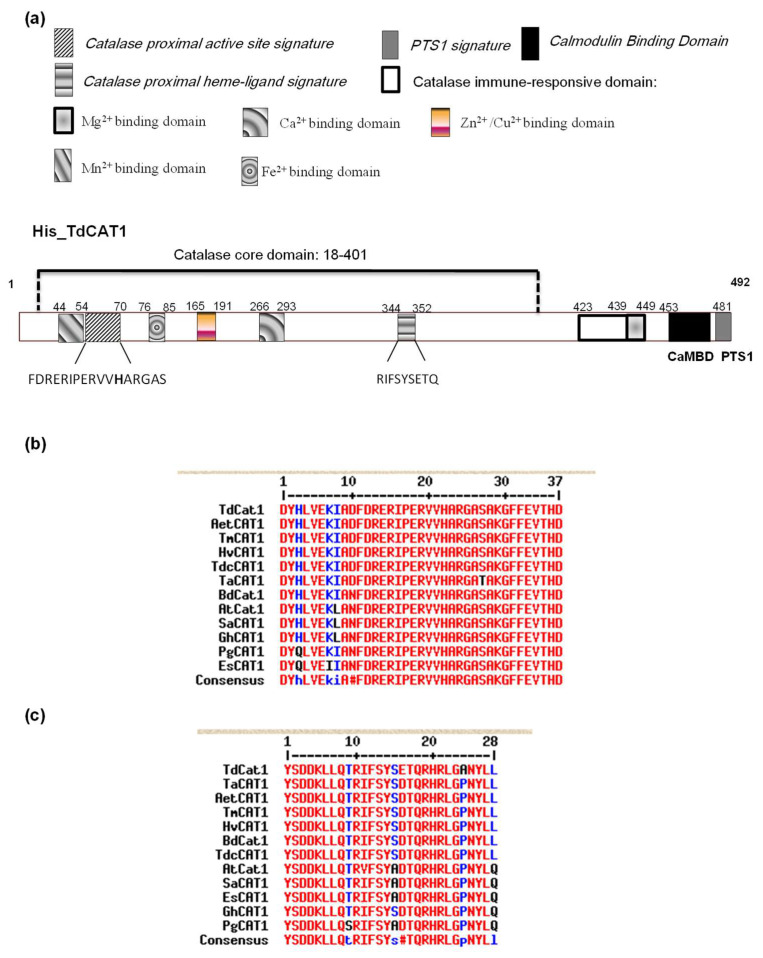
Schematic representation of TdCAT1 protein structure and alignment of the putative domains. (**a**) Bioinformatic analysis reveals the presence of different conserved domains; Catalase proximal active site signature (54–70), Catalase proximal heme-ligand signature (344–352); Catalase core domain (18–401) and catalase immune-responsive domain (423–481). (**b**) Protein sequence alignment of catalase proximal active site signature domain of TdCAT1 with other plant catalase proteins. (**c**) Protein sequence alignment of catalase proximal heme-ligand signature of TdCAT1 with other plant catalase proteins.

**Figure 2 antioxidants-11-01820-f002:**
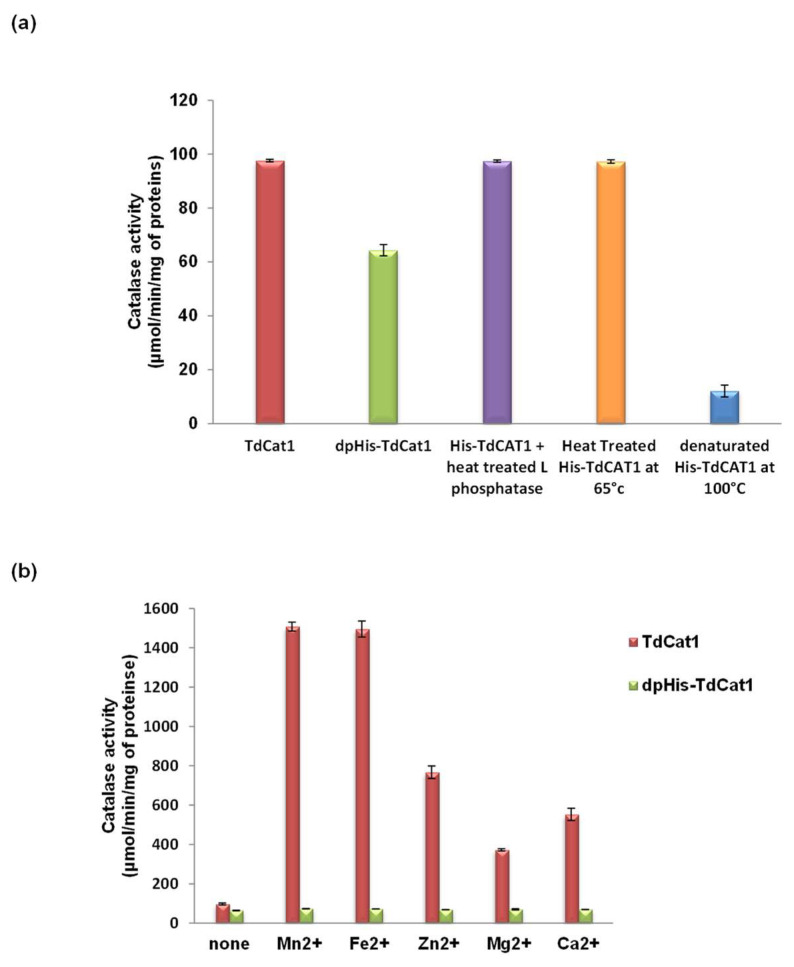
Importance of TdCAT1 phosphorylation for its catalase activity. (**a**) Catalase activity of TdCAT1 decreased after λ-phosphatase treatment. (**b**) Addition of divalent cations did not affect the catalytic activity of TdCAT1. Indicates value significantly different from the control. Statistical significance was assessed by applying the student *t*-test at *p* < 0.01.

**Figure 3 antioxidants-11-01820-f003:**
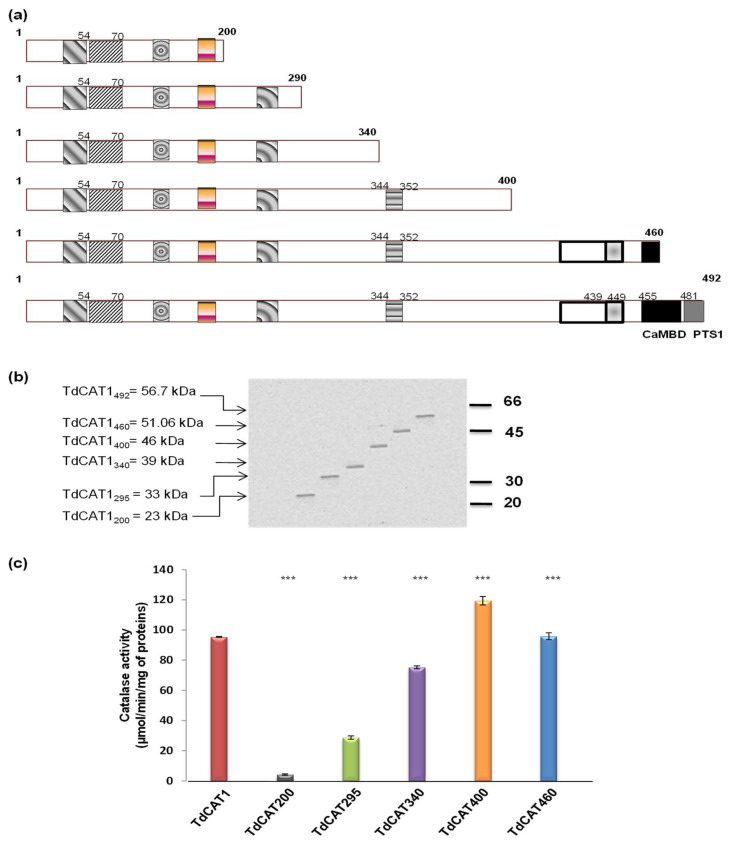
TdCAT1 harbors a putative autoinhibitory domain in its C-terminal portion. (**a**) Schematic representation of different deleted forms used in analyzing catalase activity of TdCAT1. (**b**) Verification of the correct size of deleted forms of TdCAT1 by Western blot using anti-Histidine antibodies. (**c**) Catalase activity of the different forms TdCAT1, TdCAT_200_, TdCAT_295_, TdCAT_340_, TdCAT_400_, and TdCAT_460_. Data are means ± SE of three biological replicates. (***) indicates value significantly different from the control. Statistical significance was assessed by applying the student *t*-test at *p* < 0.01.

**Figure 4 antioxidants-11-01820-f004:**
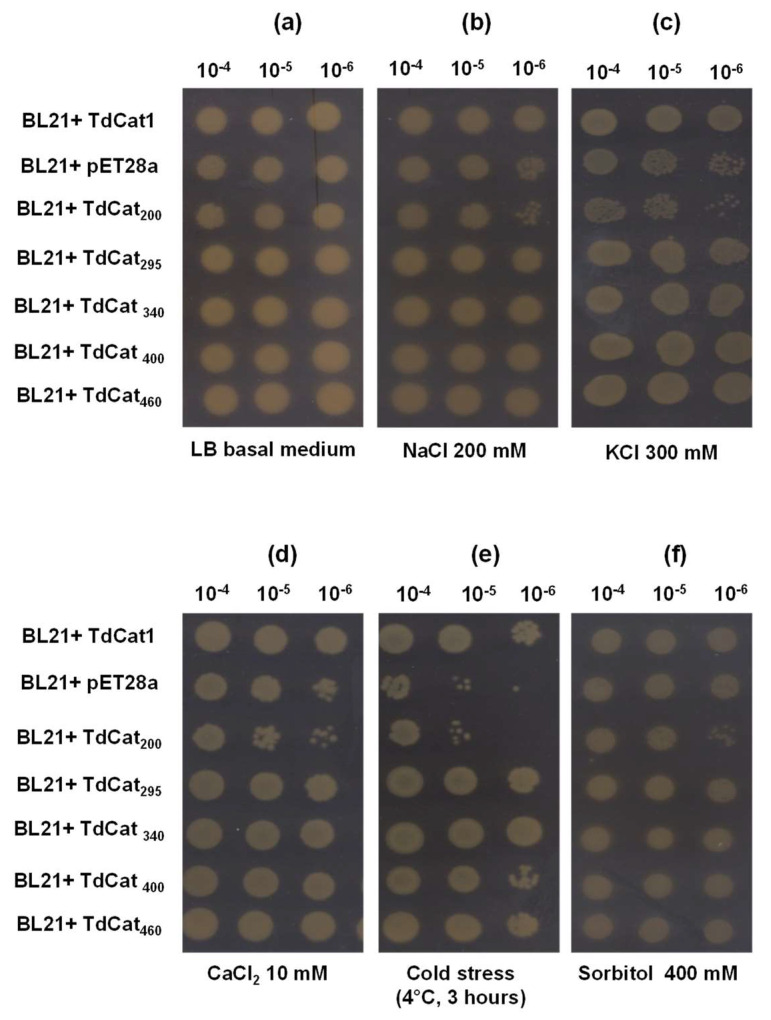
Behavior of TdCAT1 and derivative-forms-transformed bacteria in liquid LB medium in the presence or absence of abiotic stress. *E. coli* cells transformed with the empty vector (pET28a) or with the recombinant vectors (pET28a + TdCAT1 or its derivatives) were grown for 24 h under normal growth conditions (LB medium) (**a**) or after the addition of 200 mM NaCl (**b**), 200 mM KCl (**c**), 10 mM CaCl_2_ (**d**), cold stress (4 °C) (**e**), or 400 mM Sorbitol (**f**). Data are means ± SE of three biological replicates.

**Figure 5 antioxidants-11-01820-f005:**
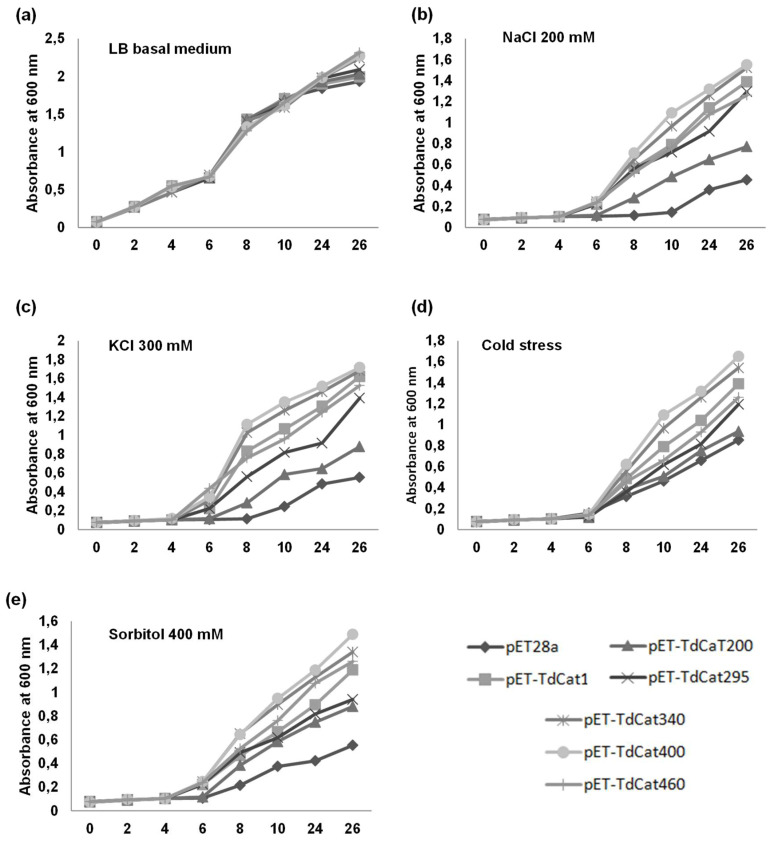
Behavior of TdCAT1 and derivative-forms-transformed bacteria in liquid LB medium in normal conditions (**a**) or in presence of 200 mM NaCl (**b**); 200 mM KCl (**c**) or cold stress (4 °C; **d**), or 400 mM Sorbitol (**e**). *E. coli* cells transformed with the empty vector (pET28a) or with the recombinant vectors (pET28a + TdCAT1 or its derivatives) were grown for 24 h in different medium. Data are means ± SE of three biological replicates.

**Figure 6 antioxidants-11-01820-f006:**
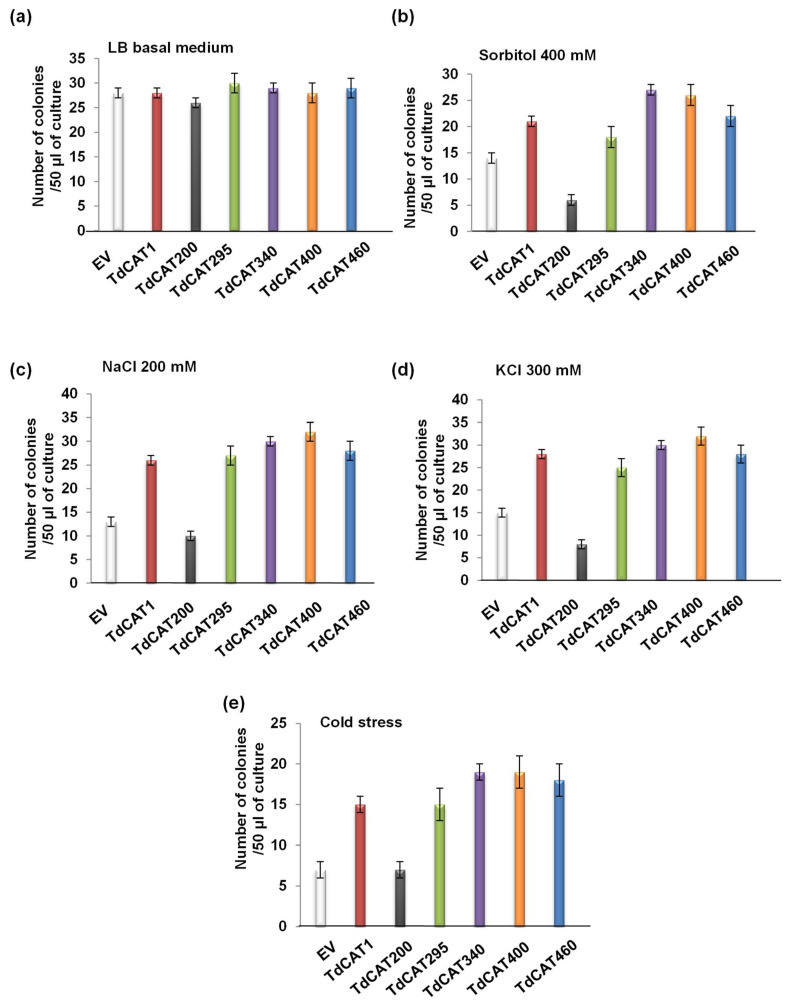
Percentage of cells’ viability under control (**a**), 400 mM Sorbitol (**b**), 200 mM NaCl (**c**), 200 mM KCl (**d**), or cold stress (4 °C) (**e**). Data presented are means of at least 3 independent experiments ± S.E. Mean values that were significantly different at *p* < 0.01 from each other are marked with asterisks indicates value significantly different from the control. Statistical significance was assessed by applying the student t-test at *p* < 0.01. Ev: Empty vector.

**Figure 7 antioxidants-11-01820-f007:**
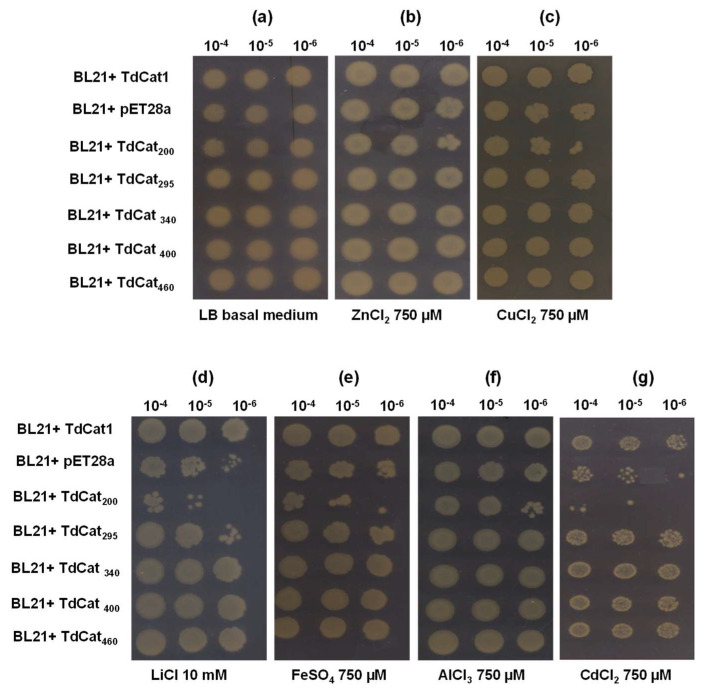
Functional characterization of TdCAT1 gene expressed in *E. coli* (BL21 strain) in response to heavy metals stress. Bacterial cells transformed with the empty vector (pET28a) and with the recombinant vectors (pET28a + TdCAT1 or derivative forms) were grown for 24 h under normal growth conditions (LB medium, (**a**)) or after the addition of 750 µL of ZnCl_2_ (**b**), 750 µM CuCl_2_ (**c**), 10 mM LiCl (**d**), 750 µM FeSO_4_ (**e**), 750 µM AlCl_3_ (**f**), or 750 µM CdCl_2_ (**g**).

**Figure 8 antioxidants-11-01820-f008:**
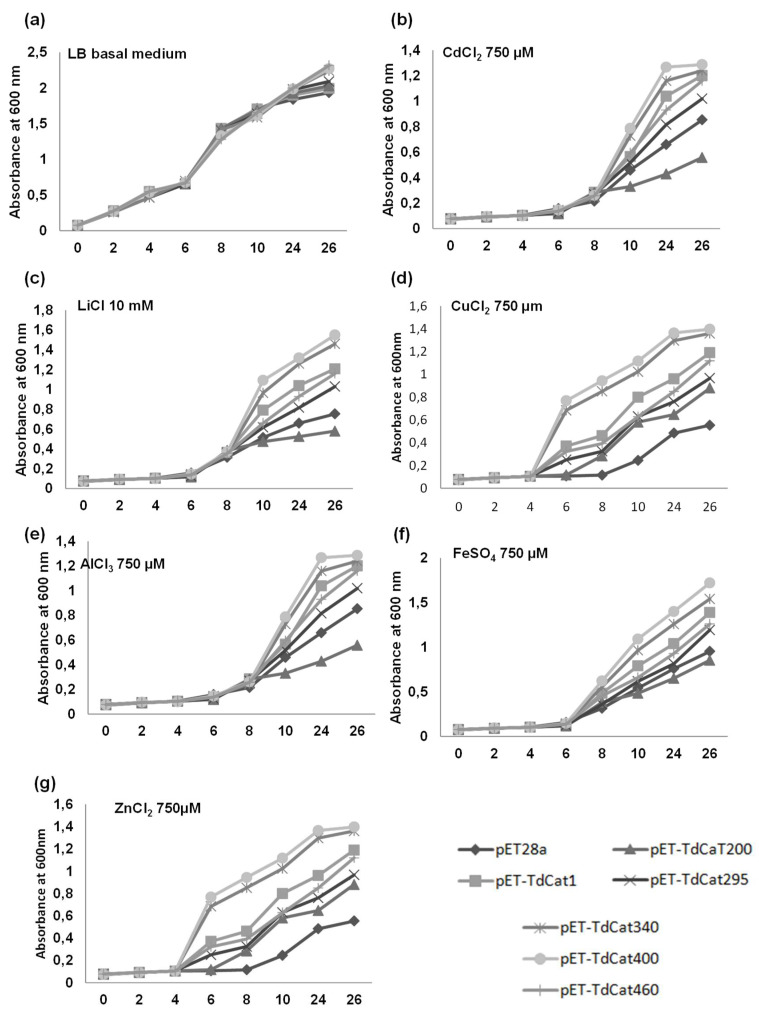
Behavior of TdCAT1 and derivative-forms-transformed bacteria in liquid LB medium in normal conditions or in the presence of metallic stress. *E. coli* cells transformed with the empty vector (pET28a) or with the recombinant vectors (pET28a + TdCAT1 or its derivatives) were grown for 24 h under normal growth conditions (LB medium, (**a**)) or after the addition of 750 µL of CdCl_2_ (**b**), 10 mM LiCl (**c**), 750 µL CuCl_2_ (**d**), 750 µL AlCl_3_ (**e**), 750 µL FeSO_4_ (**f**), or 750 µL ZnCl_2_ (**g**). Data are means ± SE of three biological replicates.

**Figure 9 antioxidants-11-01820-f009:**
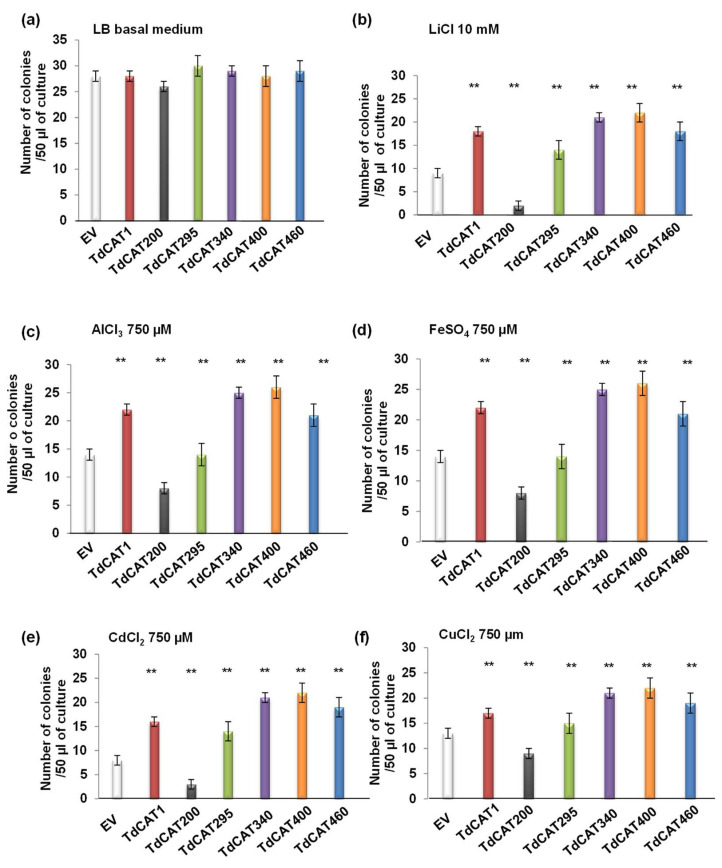
Percentage of cells’ viability under control and heavy metal stress conditions. *E. coli* cells transformed with the empty vector (pET28a) or with the recombinant vectors (pET28a + TdCAT1 or its derivatives) were grown for 24 h under normal growth conditions (LB medium, (**a**)) or after the addition of 10 mM LiCl (**b**), 750 µL AlCl_3_ (**c**), 750 µL FeSO_4_ (**d**), 750 µL of CdCl_2_ (**e**), or 750 µL CuCl_2_ (**f**), Ev: Empty vector. Data presented are means of at least three independent experiments ± S.E. Mean values that were significantly different at *p* < 0.01 from each other are marked with asterisks (**). (**) indicates value significantly different from the control. Statistical significance was assessed by applying the student t-test at *p* < 0.01.

## Data Availability

Data are contained within the article or Appendix A.

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
