# Peer review of "The Putative Auto-Inhibitory Domain of Durum Wheat Catalase (TdCAT1) Positively Regulates Bacteria Cells in Response to Different Stress Conditions"

_antioxidants, 2022, doi:10.3390/antiox11091820_

Round 1

Reviewer 1 Report

The paper by Ghorbel et al. contains an interesting piece of information dealing on the antioxidant properties of a catalase protein previously isolated from durum wheat. The enzyme structure has been clearly described and so the effect of some protein modifications (phosphorylation) and the putative role of some of its motifs. Despite some weakness in the style of presentation (particularly excessive reiteration of some statements, making the text at times "redundant", and language inaccuracies), it is suggested to accept the paper after relatively minor revisions. To help the revision process, a file containing several suggested modifications is attached. It is advised to consider such suggestions, aimed at making the message more accurate and clearer for the reader's sake. It is also recommended to check the correct writing and numbering of References (some possible mistakes have been noted).

Author Response

  • We thank the reviewer for his precious remarks. We considered all suggestions (we deleted redundant sentences, we verified the references and we checked the spelling).

Reviewer 2 Report

The authors characterized a wheat catalase by transgenic expression in bacteria and screening for activity. They found variable activity in the expressed fragments. The knowledge will be useful for studies in this field. 

The authors claim that " the presence of 4 different S-Nitrolysated residues was described here. (line  429)". But the only instance of explaining their results in this regard was stated "In an other hand, we founded that TdCAT1 presents four different S-nitrosated cysteine in its sequence (starting line - 364_.) without mentioning how these conclusions were obtained. The authors should clearly describe how they found these nitrosylated residues, correctly spelling "nitrosylated" in each usage of the word. 

Author Response

  • We thank the reviewer for his comment and the word “nitrosylated” was verified all over the text. Moreover, the nitrosylated residues were predicted using the GPS-SNO 1.0 database

“We also added the following paragraph to the texte: L391-L398: ’’On the other hand, we found that the TdCAT1 sequence presents four different S-nitrosylated cysteine residues as revealed by the GPS-SNO 1.0 database. S-nitrosylation controls reactive nitrogen species (RNS) and ROS homeostasis in plants via catalase proteins.  S-nitrosylation is regulated by a goup of proteins termed transnitrosylase [34]. Recently, it has been demonstrated that Arabidopsis CAT3 is required for S-nitrosylation. CAT3 presented a transnitrosylase activity at Cys-10 [35].  This activity is regulated by a highly conserved residue (Cyc-343) wich is not conserved in CAT2 (the enzyme responsable for the majority of catalase activity in Arabidopsis [35]. »

Reviewer 3 Report

In this manuscript, the authors identify through bioinformatic analyses that the catalase TdCAT1 from durum wheat contains different novel conserved domains. They purify the protein from E. coli and demonstrate that various mutants with deleted domains exhibit altered catalase dynamics. They then illustrate that expression of TdCAT1 in E. coli results in a a variety of growth defects under different growth conditions.

The bioinformatic and in vitro catalase portions of the paper seem well done and the conclusions drawn are supported by the evidence. However, I do not understand the motivation or relevance of the effects of TdCAT1 expression on E. coli growth. E. coli BL21(DE3) generally expresses extremely high levels of exogenous proteins; indeed it is for this reason that it is used as an expression vehicle for protein purification, as the authors use it here. It is not at all unusual or unexpected for high expression levels of exogenous protein to affect the growth dynamics of E. coli, and it is not motivated within the paper why the effects of a wheat protein on the growth of enteric bacteria is of any interest.

In addition, there are throughout a variety of typos and English language errors. I include an annotated pdf marking errors I identified, although there are likely more.

Author Response

  • We thank the reviewer for his comments. In fact, the goals of this paper are to know if the TdCAT1 can improve tolerance to abiotic stress. We started to do this in vitro study in a bacterial system (the E. coli model) before going in planta. In addition, different domains of the TdCAT 1 protein were tested based on the in siico study. The E. coli system is usually used to test the capacity of a protein to respond to different stress conditions.
  • The English language mistakes were corrected in the manuscript.

Round 2

Reviewer 3 Report

The authors have improved the presentation of the manuscript. While it is still unclear to me how testing the ability of how TdCAT1 affects the growth of E. Coli relates to plant biology, I think the authors have addressed the issue about as well as is possible.